A specimen of Paralycoptera Chang & Chou 1977 (Teleostei: Osteoglossoidei) from Hong Kong (China) with a potential Late Jurassic age that extends the temporal and geographical range of the genus

Tse Tze-Kei 1
Pittman Michael 1 mpittman@hku.hk
Chang Mee-mann 2
1 Vertebrate Palaeontology Laboratory, Life and Planetary Evolution Research Group, Department of Earth Sciences, The University of Hong Kong , Pokfulam, Hong Kong , China
2 Key Laboratory of Vertebrate Evolution and Human Origin of Chinese Academy of Sciences, Institute of Vertebrate Paleontology & Paleoanthropology, Chinese Academy of Sciences , Beijing , China
Esteban María Ángeles
Electronic publication date: 2015 Mar 26
Publication date: 2015
Volume: 3
Electronic Location ID: e865
Received 2014 Dec 18; Accepted 2015 Mar 9
Copyright: © 2015 Tse et al.
Copyright year: 2015
Copyright holder: Tse et al.
License: This is an open access article distributed under the terms of the Creative Commons Attribution License, which permits unrestricted use, distribution, reproduction and adaptation in any medium and for any purpose provided that it is properly attributed. For attribution, the original author(s), title, publication source (PeerJ) and either DOI or URL of the article must be cited.
License URL: https://creativecommons.org/licenses/by/4.0/

Keywords: Jurassic, Fish, Osteoglossomorph, Lacustrine, Volcanism, Paralycoptera, Hong Kong, Lai Chi Chong Formation

Funding: The Faculty of Science and Department of Earth Sciences of the University of Hong Kong This study was supported by a Summer Research Fellowship awarded to Tze-Kei Tse by the Faculty of Science of the University of Hong Kong. The stereo microscope and photographic equipment used were purchased with equipment funds awarded to Michael Pittman by the Faculty of Science and the Department of Earth Sciences. The funders had no role in study design, data collection and analysis, decision to publish, or preparation of the manuscript.

==============================
We describe a Mesozoic fish Paralycoptera sp. (Teleostei: Osteoglossoidei), on the basis of a postcranial skeleton collected from the volcaniclastic mudstones of the Lai Chi Chong Formation of Hong Kong, China. The new finding—representing the city’s first Mesozoic fish—extends the geographical distribution of Paralycoptera from eastern mainland China into Hong Kong, demonstrating a wider distribution than previously appreciated for this genus. A radiometric age for the Lai Chi Chong Formation of 146.6 ± 0.2 Ma implies a temporal range expansion for Paralycoptera of approximately 40 million years back from the Early Cretaceous (∼110 Ma). However, spores found in the Formation suggest an Early Cretaceous age that is consistent with the existing age assignment to Paralycoptera. We argue that the proposed temporal range extension is genuine because it is based on recent high precision radiometric age data, but given the discrepancies with the biostratigraphic ages further investigation is needed to confirm this. This study provides an important step towards revealing Hong Kong’s Mesozoic vertebrate fauna and understanding its relationship to well-studied mainland Chinese ones.

Introduction

In the summer of 2013, a fish fossil—SHGM L275—labelled as a plant was discovered in the collections of the Stephen Hui Geological Museum (SHGM) at the University of Hong Kong (HKU). The fossil (∼2 cm long) is hosted within a small mudstone fragment (5 cm by 3 cm) that was supposedly collected from the Lai Chi Chong Formation (荔 枝 莊組) of Lai Chi Chong, Tolo Channel, north-eastern New Territories, Hong Kong (Fig. 1). This provenance information is based on the specimen label, which appears to be correct, given that all fossils with the same catalogue number are lithologically similar and match the locality’s expected lithologies (see ‘Discussion’). It is not mentioned in the literature—probably because of its incorrect specimen label—unlike a fossil fish specimen from other Lantau Volcanic Group rocks in Shek Pik (石壁), Lantau Island (Campbell et al., 2007), which has a passing mention in Lee, Chan & Ho (1997) (Fig. 1). The latter specimen is supposed to be in the SHGM collections, but as it could not be located, it is assumed to have been lost. A fossil fish is also known from the summit area of Lantau Peak (鳳凰山), Lantau Island (CM Lee, pers. comm. 2014) (Fig. 1), but the exact location of this discovery is unknown. Thus, this specimen could belong to the Early Cretaceous Kau Sai Chau or Repulse Bay Volcanic Groups (141.1 ± 0.2 Ma and 142.8 ± 0.2 Ma respectively) because these rocks outcrop in the upper and lower parts of Lantau Peak respectively (Campbell et al., 2007). The fish fossil assemblages of Lai Chi Chong and Lantau Island are therefore important to compare, but the lack of available specimens from the latter currently prevents this comparison.

Figure 1 The location of the Lai Chi Chong Formation in Tolo Channel, Hong Kong.

A map of Hong Kong showing the location of the Lai Chi Chong Formation in Tolo Channel, as well as the broader Lantau Volcanic Group. The approximate locations of fossil fish discoveries in Hong Kong are marked in red. Note that Lantau Peak is now considered to belong to the Kau Sai Chau and Repulse Bay Volcanic groups (Campbell et al., 2007). Scale bar = 10 km (modified from Sewell et al., 2000).

The plant fossils discovered from the Lai Chi Chong Formation e.g., Cyathidites, Classopollis and Cicatricosisporites, suggest an Early Cretaceous age for the fossil beds (Lee, Chan & Ho, 1997). However, high precision U-Pb single crystal zircon dating of coarse crystal tuff from the upper Lai Chi Chong Formation suggests that the Formation is 146.6 ± 0.2 million years old, which corresponds to the Tithonian stage of the Late Jurassic (Campbell et al., 2007), some 40 million years earlier. The high sampling and analytical standards applied to obtaining the radiometric age for the Lai Chi Chong Formation (see Campbell et al., 2007 for details) suggests that its numerical age is unequivocal and that the plant fossil evidence deserves further detailed investigation.

This study focuses on the identification of SHGM L275 and understanding its ecology in the context of the palaeoenvironment of the Lai Chi Chong Formation, that has been inferred from its geology and plant fauna (Lee, Chan & Ho, 1997) (Fig. 2).

Figure 2 A simplified geological field sketch of Lai Chi Chong, Tolo Channel, Hong Kong.

A simplified geological field sketch of the type locality of the Lai Chi Chong Formation at Lai Chi Chong, Tolo Channel, NE New Territories, Hong Kong (simplified from Workman, 1991).

Methods and Materials

The studied fossil specimen, SHGM L275, is a partially-preserved articulated bony fish skeleton that is missing its anterior portion (Fig. 3). The specimen shows the dorsal, anal and caudal fins and is preserved in a laminated, non-fissile, pale grey orange-spotted mudstone from the Lai Chi Chong Formation (Fig. 3). SHGM L275 is now deposited in the collections of the Stephen Hui Geological Museum (SHGM) at the University of Hong Kong. The specimen was prepared mechanically using a thin needle and was examined under a Leica S8APO stereomicroscope (Leica, Weltzar, Germany) which has a magnification range of 10–80×. Photographs were taken of the specimen using a range of focal points with a Nikon D610 DSLR camera mounted to the stereomicroscope. The images were uploaded into the open-access computer software CombineZP (www.combinezp.software.informer.com/) to focus-stack them into fully-focused composite images. Based on a preliminary comparative study against Nelson (2006), SHGM L275 was diagnosed as an osteoglossomorph fish based on the possible presence of an epural and 15 principal branched caudal fin rays (Xu & Chang, 2009). The specimen was then compared by standard methods with other Chinese Mesozoic osteoglossomorph fish from the collections of the Institute Vertebrate Paleontology and Paleoanthropology (IVPP; Beijing, China) and the Stephen Hui Geological Museum (SHGM (HKU), Hong Kong) (see Table S1 in the Supplemental Information). The specimen’s features were then coded against character lists from osteoglossomorph-specific phylogenetic analyses (Shen, 1996; Zhang, 2006; Li & Wilson, 1996; Wilson & Murray, 2008; Xu & Chang, 2009). The review of the osteoglossoid osteoglossomorph Paralycoptera by Xu & Chang (2009) was particularly important towards the referral of SHGM L275 to this genus because of its details on anatomical variation.

Figure 3 Specimen SHGM L275.

(A) Magnified image (10.5x) of the specimen SHGM L275. The circular features in the anterior portion of the specimen appear to be the anterior rims of vertebrae. Identified vertebrae are numbered from 1 to 20, with 1 being an abdominal vertebra, and the remaining (19)—caudal vertebrae. Scale bar = 1 cm. (B) Specimen SHGM L275 before further preparation. Scale bar = 1 cm.

Results

The specimen SHGM L275 represents a small fish. The preserved part corresponds to the caudal portion of the fish, with the head and abdominal portion missing. The total length of the preserved part, including the caudal fin, is approximately 18 mm. Twenty vertebrae are identified (Fig. 3) between the anterior part of the dorsal and anal fins and the caudal fin, with the anteriormost being an abdominal vertebra, and the remaining (19)—caudal vertebrae. The number of caudal vertebrae is comparable to many stem osteoglossomorphs, like Huashia gracilis and Jinanichthys longicephalus (Wilson & Murray, 2008). Most of the vertebral centra are dorsoventrally deeper than anteroposteriorly long, which may allow easier lateral movements during propulsion, as in most fishes. In the anterior part of the specimen, there are four circular features directly on the vertebral column (Fig. 3)—these are the anterior rims of the vertebrae. This feature is also present in the osteoglossoid osteoglossomorph Paralycoptera wui, IVPP V2989.100 (Fig. 4), and in other studied osteoglossomorph specimens like Yanbiania wangqingica, IVPP V6767-1, and Tongxinichthys microdus, IVPP 2332.1 (Wilson & Murray, 2008).

Figure 4 Paralycoptera, IVPP V2989.100, reveals numerous circular vertebral rims.

Paralycoptera, IVPP V2989.100, has a partially disarticulated vertebral column that reveals numerous circular vertebral rims (most of them are impressions), as in SHGM L275. Scale bar = 1 cm.

In SHGM L275, the anal fin is larger than the dorsal fin like those in Paralycoptera wui (Chang & Chou, 1977; Xu & Chang, 2009). Seventeen fin rays were observed in the anal fin whilst 10 were observed in the dorsal fin, although the actual number of fin rays may be higher because the anterior ends of both fins are incomplete (Fig. 5). However, the fin ray counts—as they are—are the same as those of Paralycoptera wui IVPP V2989.100 and .105, although the fins of the latter specimen are also incomplete, as in SHGM L275. For both the anal and dorsal fins, the lengths of the fin rays are longer in the anterior portion of the fin than in the posterior portion giving them a sub-triangular shape. The preserved anterior margins of the anal and dorsal fins are opposite to each other and are rather close to the caudal fin suggesting that the dorsal fin is posteriorly situated along the fish. Such features, together with the shape of the fins, are seemingly similar to the posterior portion of Lycoptera, but in the latter taxon the size difference between the anal and dorsal fin is not significant compared to SHGM L275. Between the fins and the vertebrae, pterygiophores supporting the fin rays are observed (Fig. 5). The number of pterygiophores is more or less the same as the number of fin rays because the ends of each pterygiophore preserved leads to the base of a fin ray.

Figure 5 The anterior portion of SHGM L275.

Magnified image (10.5×) of the anterior portion of SHGM L275 showing the position of the fin rays in the anal and dorsal fins. Scale bar = 0.5 cm.

In the caudal skeleton of SHGM L275 (Fig. 6), six hypurals were identified. The first one is posteriorly broader, giving a fan-like shape, whereas the second is comparatively narrow. These hypurals articulate with the first ural centrum, and support the rays of the lower lobe of the caudal fin. Under the first hypural, the parhypural, articulating with the preural centrum 1, also has a somewhat fan-shaped broader posterior portion. The second ural centrum is triangular in shape and is slightly upturned towards the upper lobe of the fin. The third to sixth hypurals are rectangular rod-shaped, articulating with the second ural centrum, and supporting the rays of the upper lobe of the caudal fin. Comparing ural centrum 2 with ural centrum 1 and neighbouring vertebral centra, ural centrum 2 is anteroposteriorly longer than dorsoventrally deep whilst the others are dorsoventrally deeper than anteroposteriorly long. In the area above the ural centra 1 and 2, traces of uroneurals can be seen, though it is difficult to estimate their number (possibly two or three). The anterior tip(s) of the uroneurals extend to the posterodorsal end of the preural centrum 1. An epural is probably present above the uroneurals. No urodermals were found.

Figure 6 The caudal skeleton and bases of caudal fin rays in SHGM L275.

Magnified image of the caudal skeleton and bases of caudal fin rays in SHGM L275, the arrows point to the outermost (unbranched) principal caudal fin rays. Abbreviations: ep, epural; h1-6, hypurals 1-6; hsp2-5, haemal spines on preural centrum 2-5; nsp1-5, neural spines on preural centrum 1-5; nspu1, neural spine on u1; ph, parhypural; pr.r, procurrent rays; pu1, preural 1; u1, u2, ural centra 1 and 2; un, uroneurals. Scale = 1 mm.

Even though the caudal fin rays are poorly-preserved, the caudal fin appears to be symmetrical because the vertebral column only bends slightly towards the upper lobe. Thus, the specimen is likely to possess a homocercal tail, which is a trait of all teleostean fish (Nelson, 2006). We were able to find out the approximate counts of the caudal fin rays: 17 principal fin rays are recognized, seven branched rays with one unbranched ray at the upper margin in the upper lobe, and eight branched rays with one unbranched ray at the lower margin in the lower lobe. Besides, about 5–6 and 3–4 short, procurrent rays are observed in front of the upper and lower lobe respectively. Five neural spines on the 1st–5th preural centra and four haemal spines under 2nd–5th preural centra are prolonged, the posterior ones of which are in support of the procurrent rays. The ural centrum 1, perhaps, also carries a short neural spine (Fig. 6).

Based on the features described above, especially that a possible epural is present, the number of branched caudal fin rays is 15 and the dorsal fin is posteriorly situated, SHGM L275 most likely belongs to the order Osteoglossiformes (Shen, 1996; Xu & Chang, 2009), under the superorder Osteoglossomorpha (Greenwood et al., 1966).

SHGM L275 was added to the osteoglossomorph phylogenetic data matrices of Shen (1996), Zhang (2006), Wilson & Murray (2008) and Xu & Chang (2009) (Table 1) and in all four analyses the taxon that has the most similar codings was Paralycoptera. However, for the Zhang (2006) matrix, SHGM L275 has more closely matched codings to Singida than to Paralycoptera. The Eocene temporal range of Singida (Xu & Chang, 2009) is at odds with the Late Jurassic age of SHGM L275, but it might be possible that the new specimen supports an extremely large range extension. However, SHGM L275 is referable to Paralycoptera based on additional details of the caudal skeleton: the two hypurals in the lower lobe of Paralycoptera are separated and unfused (Shen, 1996) like in SHGM L275, whilst those in Singida are partially fused (Murray & Wilson, 2005). In addition, Singida has a falcate anal fin instead of the triangular one in Paralycoptera (Murray & Wilson, 2005) and SHGM L275.

Table 1 Phylogenetic coding similarities between SHGM L275 and Paralycoptera.

The applicable characters from Shen (1996), Zhang (2006), Wilson & Murray (2008) and Xu & Chang (2009) to SHGM L275: coding similarities with the most closely-matched genus—Paralycoptera (○, matched; x, not matched). For the codings of the individual studies please see Tables S2–S5 in the Supplemental Information).

Osteoglossomorph study	Equivalent character numbers	
Shen (1996)	28	29	33	34	35	36		
Zhang (2006)	47	48	49	53	54	61	60	
Wilson & Murray (2008)	69	67	68	71		65	64	
Xu & Chang (2009)	54	55	56			62		
SHGM L275 compared to Paralycoptera	○	x	x	○	○	○	○	

In comparing SHGM L275 and Paralycoptera based on the above analyses (Table 1 and S2–S5), there were a few character state discrepancies. These unmatched characters include: (1) the condition of the neural spine on ural centrum 1—whether the neural spine is complete or rudimentary, and (2) the number of epurals. According to Wilson & Murray (2008), the neural spine on the first ural centrum of Paralycoptera should be absent or rudimentary, whereas Shen (1996) and Xu & Chang (2009) observe a completely developed neural spine. Zhang (2006) is uncertain about the relative development of this spine, but in SHGM L275 a rudimentary neural spine is present. The number of epurals present in Paralycoptera remains controversial. Shen (1996) identified a single epural in Paralycoptera whereas Xu & Chang (2009) noted its absence. In specimens IVPP V2989.65, .100 and .105 of Paralycoptera, we also identified no epurals, like Xu & Chang (2009). An ‘x’ has been marked in Table 1 for this character, even though the character state used by Xu & Chang (2009)—‘one or absent’—should justify the use of a ‘○’ mark instead. We therefore advocate the separation of this state in future work in accordance with Greenwood (1970) and the epural characters of Shen (1996), Zhang (2006) and Wilson & Murray (2008). There is a possible epural in SHGM L275. Zhang (2006) and Wilson & Murray (2008) both record uncertainty in the number of epurals in Paralycoptera. The first preural centrum of SHGM L275 has a complete neural spine, as identified in Paralycoptera by all four aforementioned analyses, but Xu & Chang (2009) mistakenly recorded a ‘rudimentary or absent’ neural spine in their data matrix. Excluding the aforementioned discrepancies, the four studies otherwise converge on SHGM L275 being a specimen of Paralycoptera. However, Xu & Chang’s (2009) observations of individual anatomical variation within Paralycoptera actually explain the differences in the caudal skeleton observed by Shen (1996), Zhang (2006) and Wilson & Murray (2008). Therefore, this confirms that SHGM L275 is a specimen of Paralycoptera (Fig. 7), which in our opinion negates the need for a numerical phylogenetic analysis. Xu & Chang (2009) synonymised the genus into one species P. wui whose features in SHGM L275 are:

(1) a completely developed neural spine on the first preural centrum;

(2) two or three uroneurals;

(3) four upper hypurals and two lower hypurals, and

(4) all hypurals are independent.

Figure 7 Skeletal reconstruction of Paralycoptera.

Reconstructed skeleton of Paralycoptera (Xu & Chang, 2009; used with the permission of the authors).

Systematic Palaeontology

SUBDIVISION TELEOSTEI Müller, 1846	
SUPERORDER OSTEOGLOSSOMORPHA Greenwood et al., 1966	
ORDER OSTEOGLOSSIFORMES Regan, 1909	
SUBORDER OSTEOGLOSSOIDEI Regan, 1909	
GENUS †PARALYCOPTERA Chang & Chou, 1977	
†PARALYCOPTERA sp.	

Discussion

Ecology of Paralycoptera

Paralycoptera is a member of both northern China’s Lycoptera Fauna and south-eastern China’s Mesoclupea Fauna (Chang & Jin, 1996). It has been discovered in Jilin, Liaoning, Shandong, Zhejiang and Fujian provinces (Xu & Chang, 2009) and now in Guangdong Province too (this study) (Fig. 8). This geographical range is impressive given that the northern part of China has been separated from the south by the Qinling-Dabie Shan orogenic belt since the Late Triassic (Hacker, Ratschbarcher & Liou, 2004), and the 20°difference in latitude between the southernmost and northernmost localities—Hong Kong, Guangdong Province and Tonghua, Jilin Province respectively—a distance of over 2,000 km. This geographic distribution may imply that Paralycoptera was adaptable to a wide range of environments compared to other members of the two faunas. However, climate variability over this geographical area was not very significant in the Late Mesozoic—climate change towards more temperate and humid conditions is reflected by geochemical weathering indices (Ohta et al., 2014) with temperatures between 5 °C and 25 °C reconstructed from oxygen isotope data from sedimentary rocks in north-eastern China (Wang et al., 2013). However, occasional semi-arid periods are indicated by the appearance of the arid plants Ephedripites and Classopollis in Hong Kong (Lee, Chan & Ho, 1997), as well as oxygen isotope data from sedimentary rocks in north-eastern China, including from Jilin and Liaoning provinces (Wang et al., 2013). Therefore, Paralycoptera most likely lived in areas with a tropical-subtropical climate similar to many modern osteoglossoids, such as Scleropages formosus (Kottelat, 2011).

Figure 8 Chinese Paralycoptera localities.

Paralycoptera localities in China (Locations from Chang & Miao, 2004).

Paralycoptera localities were all continental basins (Fig. 9) where fluvial or lacustrine deposits dominated (Chang & Jin, 1996) and these have similar lithologies (see Table 2). Vigorous tectonic activity and episodes of volcanism were common in these localities during the late Mesozoic (Chang & Jin, 1996; Chang & Chou, 1977; Li & Li, 2007). The Lai Chi Chong Formation of Hong Kong consists of mainly tuff and tuffaceous sedimentary rocks (Strange, Shaw & Addison, 1990). A shallow freshwater lake environment subject to the influence of volcanic activity is indicated by fluvial-lacustrine and volcaniclastic sedimentary facies (Strange, Shaw & Addison, 1990; Workman, 1991; Sewell et al., 2000), predominantly turbidites (Lai, Campbell & Shaw, 1996), and the discovery of terrestrial freshwater plant fossils including Equisetites, Cladophlebis exiliformis, Gleichenites gladiatus and Carpolithus (Lee, Chan & Ho, 1997). According to Lin & Lee (2012), the ‘parallel laminated fine sandstone and mudstone’ facies is the most likely origin of SHGM L275 as the only light-coloured mudstone unit is confined to this facies (grey volcaniclastic mudstone from the western portion of the Lai Chi Chong locaility; Fig. 2). This facies contains fine-grained, cross-laminated, white and grey coloured mudstone representing a depositional environment below the wave base, where suspension currents might affect deposition (Lin & Lee, 2012). The similarities in the palaeoenvironment between Lai Chi Chong and existing Paralycoptera localities (Workman, 1991; Chang et al., 2008; Hu et al., 2012; Chen, 1983a) provides additional support for the inference that Paralycoptera from Lai Chi Chong lived in shallow freshwater lakes near areas of active volcanism. One potential hypothesis to explain the association of Paralycoptera discovery sites with volcanism is that Paralycoptera may have thrived on the higher nutrient levels in the lake caused by the influx of volcaniclastic material, and/or the warmer water temperatures provided by thermo-tectonic activities. The sedimentary rocks preserved at Lai Chi Chong frequently show syn-sedimentary structures including microfaults, slumps, convolute bedding, load and flame structures, suggesting the occurrence of mass flows that might have been triggered by episodic volcanic and seismic activity directly related to the local subduction tectonic setting (Sewell et al., 2000; Campbell & Shaw, 2002). This implies that the habitat of Paralycoptera was subjected to episodic catastrophic events and was not a prolonged quiet, tranquil water body. This habitat is possibly similar to the turbid and swift-water habitat of Hiodontiformes—a closely related group to Osteoglossiformes (Gray, 1988). These episodic conditions could indicate that Paralycoptera had a high tolerance to environmental stress (highly variable sediment and nutrient input and possible changes in water temperature). However, the association of the fish with volcanism may more simply reflect the higher fossil preservation potential by volcaniclastic sediments, especially given that only one specimen is known among the strata so far. Crucially, the laminated mudstone that SHGM L275 is preserved in represents a relatively stable rather than unstable depositional setting. This also fits the living environments of most modern osteoglossoid fish which tend to prefer still water bodies e.g., Pantodon buchholzi and Scleropages formosus (Moelants, 2010; Kottelat, 2011). It therefore seems more plausible that Paralycoptera lived in relatively stable water body like their modern counterparts and probably migrated in times of environmental stress (no evidence of mass fish mortality in the rocks showing synsedimentary structures).

Figure 9 Southeast Asian Jurassic fish localities and the localities of Paralycoptera.

Jurassic fish localities in SE Asia and the localities of Paralycoptera (Modified from Chang & Miao, 2004).

Table 2 The lithology of Paralycoptera-bearing formations.

Lithological characteristics of the formations preserving Paralycoptera.

Formation	Province	Age	Major lithology (those that yield Paralycoptera are in bold font)	
Lai Chi Chong	Guangdong	146.6 ± 0.2 Ma, Tithonian, Late Jurassic (Campbell et al., 2007)	Light grey thickly laminated tuffaceous mudstone, massive black cherty mudstone, alternatively light and dark thickly laminated and cross-bedded coarse sandstone, conglomerate; greenish grey fine ash crystal tuff and rhyolite (Strange, Shaw & Addison, 1990; Lai, Campbell & Shaw, 1996; Lin & Lee, 2012)	
Fenshuiling	Shandong	Late Jurassic to Early Cretaceous (Li, 1998)	Mudstone, shale, siltstone, sandstone, conglomerate and tuff (Wang, 1985)	
Guantou	Zhejiang	∼ 110 Ma, Early Cretaceous (Xu & Chang, 2009)	Purplish grey, greyish green and greyish yellow tuffaceous siltstone, dark grey mudstone, purple sandstone; andesite and tuff breccia (Chen, 1983b; Hu et al., 2012)	
Hengtongshan	Jilin	Early Cretaceous (Han et al., 2013)	Black mudstone, oil shale and tuffite (Han et al., 2013)	
Baiyashan	Fujian	Early Cretaceous (Zhang, 2009)	Purplish red conglomerate, siltstone and sandstone (Zhang, 2009)	

Geographical distribution of Paralycoptera and the biogeography of the Mesoclupea Fauna

The discovery of Paralycoptera in Hong Kong extends the geographical distribution (Fig. 8) of the genus ∼700 km further south of the previously most southerly locality in the Baiyashan Formation of Fujian Province (Xu & Chang, 2009). This implies that Paralycoptera was much more widespread than previously thought and suggests that the genus may also be present in other similarly-aged lacustrine deposits in southeastern China (Fig. 9). Paralycoptera is a typical member of the Mesoclupea Fauna (Chang & Jin, 1996) so it is possible that the other members of this fauna such as Mesoclupea, Sinamia and Paraclupea could be found in Hong Kong in the future.

Age of the Lai Chi Chong Formation and osteoglossomorph evolution and biogeography

Another implication of SHGM L275 arises from the age of the Lai Chi Chong Formation. A Jurassic age was originally proposed by Workman (1991) based on the identification of the fossil plants Cladophlebis and Equisetites. However, subsequent studies of spore fossils (including Cicatricosisporites, Klukisporites, Cyaathidites, Classopollis and Pinuspollenite) from exposures of the Formation at Cheung Sheung (嶂上)—∼2.5 km south of Lai Chi Chong (Fig. 1)—suggest that the Formation was deposited between the Valanginian to Barremian stages of the Early Cretaceous (Lee, Chan & Ho, 1997) (Table S6). This age determination is closer to the Aptian age of other Paralycoptera specimens found elsewhere in China, based on absolute dating of volcanic units (Xu & Chang, 2009) (Table 2). However, as mentioned in the introduction (see ‘Introduction’), an Early Cretaceous age is not corroborated by the Late Jurassic radiometric age of the Formation (Campbell et al., 2007: 146.6 ± 0.2 Ma). The Lai Chi Chong Formation is stratigraphically overlain by the Long Harbour Formation (Sewell, Tang & Campbell, 2012) which has also been precisely dated at 142.8 ± 0.2 Ma, within the Berriasian stage of the Early Cretaceous (Davis, Sewell & Campbell, 1997). This gives additional support to the accuracy and reliability of the radiometric ages, which imply that SHGM L275 dates to the Tithonian stage of the Late Jurassic. However, as this age differs from the biostratigraphic ones, the latter warrants further investigation. Thus, a Late Jurassic age is cautiously assigned to SHGM L275 pending further biostratigraphic studies and the discovery of an in situ specimen—the formation and locality information of SHGM L275 are based on its specimen label, but this could not be verified with the fossil’s discoverer because they are not known. This conclusion has a profound impact on the origins of osteoglossomorphs, as it means that Paralycoptera was contemporaneous with other Late Jurassic taxa, including Lycoptera, Tongxinichthys and Anaethalion from the Lycoptera Fauna and Sinamia and Ikechaoamia from the Mesoclupea Fauna (Chang & Jin, 1996). Future ecological investigations of these faunas would therefore be valuable towards our understanding of osteoglossomorph origins, especially given the relatively disparate phylogenetic relationships of some of the aforementioned taxa e.g., Paralycoptera and Lycoptera (Xu & Chang, 2009).

Given the freshwater habitats of osteoglossomorphs, migration across an oceanic barrier was unlikely, so these fish should have a Pangean origin (Xu & Chang, 2009). However, the location of their origins, whether in Africa or Asia, has been debated (Wilson & Murray, 2008; Xu & Chang, 2009). The Late Jurassic occurrence of Paralycoptera in Hong Kong provides additional evidence to support the hypothesis (Xu & Chang, 2009) that osteoglossomorphs originated from eastern Asia, as the oldest represenatives of this clade are all known from the Late Jurassic of China, e.g., Lycoptera and Tongxinichthys (Chang & Jin, 1996) instead of Africa, which instead has members with more derived anatomical traits (Xu & Chang, 2009).

New phylogenetic characters

In our study a numerical phylogenetic analysis was not performed because existing data made it possible to assign SHGM L275 to Paralycoptera. However, in the course of this study it was noted that Singida was not easily distinguishable from Paralycoptera on the basis of existing characters relating to the posterior skeleton. Anatomical characteristics such as the degree of fusion in the hypurals and the shape of the anal fin that were not included in exisiting phylogenetic character lists would therefore be useful to include in future phylogenies:

- Hypurals in the lower lobe: [0] = independent; [1] = partially fused; [2] = fully fused.

- Anal fin shape: [0] = triangular; [1] = falcate.

Limitations and future work

The taxonomic identification of SHGM L275 was difficult because the fossil is incomplete, and is the only specimen of its kind from Hong Kong. Thus, further discoveries of Paralycoptera in the city (in the Tolo Channel area and on Lantau Island) would help to facilitate further anatomical comparisons with mainland Chinese specimens, providing additional insights into anatomical variation in this taxon (after Xu & Chang (2009)). To resolve the current discrepancies between the biostratigraphic and radiometric ages of the Lai Chi Chong Formation, and confirm the proposed temporal range extension for Paralycoptera, a reappraisal of current biostratigraphic evidence is required. Radiometric dating of fossil-bearing strata within the Formation will be particularly valuable, if suitable rocks can be identified in the future. A detrital zircon age of the matrix of SHGM L275, as well as of the ‘parallel laminated fine sandstone and mudstone’ facies it was assigned to (Lin & Lee (2012) facies scheme), would also lend supporting evidence to the age assignment. However, these three aspects are beyond the scope of this paper to address further. More detailed petrological analysis of the matrix of SHGM L275 using scanning electron microscopy would be valuable for corroborating its facies assignment and facilitating comparisons with the sedimentary facies of other Paralycoptera localities in mainland China, such as in Liaoning province (Chen, 1983a). These facies investigations, in addition to comparisons between the floras at these different localities, will be important towards elucidating the palaeoenvironment of Paralycoptera (and its co-inhabitants) in greater detail, particularly in relation to neighbouring volcanic activity. Future fossil collection and petrological analysis of non-Lai Chi Chong Formation Lantau Volcanic Group sediments—such as those on Lantau Island—as well as sediments from the Kau Sai Chau and Repulse Bay Volcanic Groups, will improve our understanding of local variations in the palaeoenvironment of Paralycoptera, and will potentially provide evidence of how this taxon (and its co-inhabitants) responded to the well-documented episodes of Middle Jurassic to Early Cretaceous volcanism in Hong Kong (Sewell et al., 2000). The latter narrative therefore makes Hong Kong an ideal place to understand the biotic response of Mesozoic fossils to significant environmental stress, so it is hoped that this will lead to further development of palaeontological studies in Hong Kong.

Conclusions

A fossil fish, SHGM L275, from Lai Chi Chong, Hong Kong was rediscovered in the fossil collections of Stephen Hui Geological Museum at the University of Hong Kong. This specimen is identified as Paralycoptera sp. based on the following four anatomical characteristics:

(1) a completely developed neural spine on the first preural centrum;

(2) two or three uroneurals;

(3) four upper hypurals and two lower hypurals, and

(4) all hypurals are independent.

The discovery of Paralycoptera in Late Jurassic-aged strata in Hong Kong—the city’s only Mesozoic vertebrate—appears to extend the temporal range of the genus back by ∼40 million years. However, discrepancies between the biostratigraphic and radiometric ages of the strata, which belongs to the Lai Chi Chong Formation, warrants a cautious treatment of the proposed temporal range extension, pending further geochronological investigation. However, our discovery unequivocally extends Paralycoptera’s geographical range approximately 700 km southwards, potentially affecting the Mesoclupea Fish Fauna. In the context of the geological literature on the Lai Chi Chong Formation and our knowledge of the fossil’s matrix, it is suspected that Paralycoptera lived in freshwater lakes in close proximity to volcanic environments that experienced episodic earthquakes and volcanic eruptions that greatly affected the lake’s regime. This palaeoenvironment appears to match those of other Paralycoptera localities in mainland China inspiring the conclusion that this taxon was potentially tolerant of high environmental stresses and may even have thrived on higher nutrient levels and changeable water temperatures in the lake, during times of volcanic activity.

This study makes an important contribution to our understanding of Hong Kong’s fossil heritage, given that the city has a relatively poor fossil record and limited sedimentary rock exposures (Lee, Chan & Ho, 1997). This study is the first on Hong Kong fossils in over 15 years (Lee, Chan & Ho, 1997) so it is hoped that it can help to promote further interest in Hong Kong’s palaeontology, particularly given the rare opportunity to study the biotic response to long-lived and accurately-dated Mesozoic volcanic events.

Supplemental Information

Supplemental Information 1 Supplementary Information, including Chinese language abstracts

Supplementary tables detailing the specimens studied (Table S1) and the phylogenetic characters—and associated codings—used to identify SHGM L275 (Table S2–S6). Chinese language abstracts in Cantonese and Mandarin are also included.

Click here for additional data file.

This study developed from TKT’s undergraduate research work at the Department of Earth Sciences of the University of Hong Kong. This comprised of a Summer Research Fellowship and his final year undergraduate project (EASC3308 Earth Sciences Project)—both of which were supervised by MP. The authors thank the Stephen Hui Geological Museum, the Department of Earth Sciences of the University of Hong Kong and the Institute of Vertebrate Paleontology and Paleoanthropology (IVPP) for loaning their specimens for this study. Mr. Wang Zhao (王釗) at the IVPP is thanked for further preparation of the specimen, and Dr. Wu Feixiang (吳飛翔) also from the IVPP is thanked for producing illustrations. We would like to thank Dr. Roderick J. Sewell (Hong Kong Geological Survey), Prof. Desui Miao (苗德歲) (University of Kansas) and an anonymous reviewer for their helpful feedback that improved the quality of this manuscript.

Additional Information and Declarations

Competing Interests

Author Contributions

The authors declare there are no competing interests.

Tze-Kei Tse performed the experiments, analyzed the data, wrote the paper, prepared figures and/or tables, reviewed drafts of the paper.

Michael Pittman conceived and designed the experiments, performed the experiments, analyzed the data, contributed reagents/materials/analysis tools, wrote the paper, prepared figures and/or tables, reviewed drafts of the paper.

Mee-mann Chang analyzed the data, contributed reagents/materials/analysis tools, wrote the paper, prepared figures and/or tables, reviewed drafts of the paper.

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
