# Peer review of "A specimen of Paralycoptera Chang & Chou 1977 (Teleostei: Osteoglossoidei) from Hong Kong (China) with a potential Late Jurassic age that extends the temporal and geographical range of the genus"

_PeerJ, doi:10.7717/peerj.865_

## Round 0.1 · original submission · Minor Revisions

· Academic Editor

Minor Revisions

Please revise all the annotations on the manuscript and improve the conclusions.

·

Basic reporting

This is the first ever report of a Mesozoic vertebrate fossil from Hongkong, and thus holds a special place in our uderstanding of the geological and biological history of that city.The report also extends the fossil fish's temporal and spatial distribution and reveals its link to the mainland China's mesozoic fish fauna as well as its relationship with the Jussic Asiatic ichthyfaunas.Thus, this represents a very important discovery and should be promptly published.

Experimental design

Although the fossil material is not very well preserved, the authors took great pains in adequately preparing,describing, and illustrating the specimen. The description is exhaustive and very careful.The illustrations are well done.

Validity of the findings

The authors drew very careful and cautious,but nevertheless informative and reasonable, conclusions from the rather fragmentary material,thus validifying the scientific importance and scholarly soundness of their findings. I have no complaints whatsoever about the scholarship of the paper.

·

Basic reporting

No Comments

Experimental design

No Comments

Validity of the findings

No Comments

Additional comments

This is an interesting paper which I enjoyed reading very much. It is well written, carefully and systematically organized, and well-illustrated. The supporting information is also very helpful. It is worthy of publication.

The main scientific contribution of the paper is the discovery and reporting of the first Mesozoic fish fossil from Hong Kong, and the potential spatial and temporal range expansion of Paralycoptera. It is unfortunate that there is no absolute certainty on where the specimen was collected, though the circumstantial evidence for the link with the Lai Chi Chong Fm locality is reasonably compelling. A detrital zircon age on a portion of the sample itself, along with a sample from the ‘parallel laminated fine sandstone and mudstone’ facies of the Lai Chi Chong Fm would have provided unequivocal evidence for a link. Nevertheless, if reference is made to the radiometrically-dated overlying stratigraphic unit, this would lend supporting evidence to the age assignment. I have annotated this on the ms.

The link with Lantau Volcanic Group on Lantau Island needs to be done cautiously since there are no available fish specimens, and most of the fossil localities are rather poorly defined. The upper part of Lantau Peak is also now considered to belong to both the Repulse Bay Volcanic Group and the Kau Sai Chau Volcanic Group (see Campbell et al. 2007) which have been radiometrically dated as 143 Ma and 141 Ma, respectively.

A couple of key references are missing. The Lai Chi Chong Fm was first defined from its type locality by Strange et al. 1990. Thus, it should replace the reference to Lai et al. 2006, which covered only a tiny island outcrop of Lai Chi Chong Fm on the southern edge of Geological Map Sheet No. 4. Reference should also be made to Campbell and Shaw (2002) who provide an excellent account of the structures in the Lai Chi Chong Fm and also give some good interpretation of the palaeodepositional environment.

I have not made comments on the palaeontology and will leave this to the experts.

Minor Comments

I have suggested minor amendments on the annotated manuscript. (see attached). There are some errors and omissions in the references, and some minor formatting issues.

Reviewer 3 ·

Basic reporting

No Comments

Experimental design

No Comments

Validity of the findings

No Comments

Additional comments

This is an interesting paper that describes a Mesozoic fish Paralycoptera sp. (Teleostei: Osteoglossoidei), on the basis of a postcranial skeleton recently collected from the volcaniclastic mudstones of the Lai Chi Chong Formation in Hong Kong, China. The new finding extends the geographical distribution of Paralycoptera from eastern Mainland China into Hong Kong, demonstrating a wider distribution than previously appreciated for this genus. The age of the fossil beds is controversial. The discovery supports an Early Cretaceous age determination for the Lai Chi Chong Formation. However, the authors believe that the radiometric data is accurate and unequivocal. This conclusion is sloppy. As a palaeontologist, you should come to a conclusion based on the fossils you found.

Annotated reviews are not available for download in order to protect the identity of reviewers who chose to remain anonymous.

---

## Round 0.2 · accepted · Accept

· Academic Editor

Accept

I hope you consider PeerJ again in future for publishing your results.